# Update on the Discovery of Efflux Pump Inhibitors against Critical Priority Gram-Negative Bacteria

**DOI:** 10.3390/antibiotics12010180

**Published:** 2023-01-15

**Authors:** Nina Compagne, Anais Vieira Da Cruz, Reinke T. Müller, Ruben C. Hartkoorn, Marion Flipo, Klaas M. Pos

**Affiliations:** 1Univ. Lille, Inserm, Institut Pasteur de Lille, U1177—Drugs and Molecules for Living Systems, F-59000 Lille, France; 2Institute of Biochemistry, Goethe-University Frankfurt, Max-von-Laue-Str. 9, D-60438 Frankfurt am Main, Germany; 3Univ. Lille, CNRS, Inserm, CHU Lille, Institut Pasteur Lille, U1019—UMR 9017—CIIL—Center for Infection and Immunity of Lille, F-59000 Lille, France

**Keywords:** efflux pump inhibitor, RND multidrug efflux pump, Gram-negative bacteria, antimicrobial resistance, antibiotic resistance breakers, AcrB, MexB

## Abstract

Antimicrobial resistance (AMR) has become a major problem in public health leading to an estimated 4.95 million deaths in 2019. The selective pressure caused by the massive and repeated use of antibiotics has led to bacterial strains that are partially or even entirely resistant to known antibiotics. AMR is caused by several mechanisms, among which the (over)expression of multidrug efflux pumps plays a central role. Multidrug efflux pumps are transmembrane transporters, naturally expressed by Gram-negative bacteria, able to extrude and confer resistance to several classes of antibiotics. Targeting them would be an effective way to revive various options for treatment. Many efflux pump inhibitors (EPIs) have been described in the literature; however, none of them have entered clinical trials to date. This review presents eight families of EPIs active against *Escherichia coli* or *Pseudomonas aeruginosa*. Structure–activity relationships, chemical synthesis, *in vitro* and *in vivo* activities, and pharmacological properties are reported. Their binding sites and their mechanisms of action are also analyzed comparatively.

## 1. Introduction

The spread of antibiotic resistance represents a massive global health problem. While, on the one hand, multiresistant or even panresistant pathogens have become reality in clinics, the potential for discovering new compound classes has been recently revived but yet is often still in preclinical stages [1]. Over the past 30 years, the development of antimicrobial agents was almost exclusively limited to the modification of existing compounds [2,3]. Teixobactin, one of the last newly discovered classes of antibiotics, is effective against Gram-positive bacteria but not against Gram-negative pathogens such as *Acinetobacter baumannii*, *Pseudomonas aeruginosa,* and certain *Enterobacteriaceae*, whose resistance situation has been classified as particularly critical by the WHO [4,5]. For the Gram-negative bacterium *Pseudomonas aeruginosa*, inhaled murepavadin, a peptidomimetic antibiotic targeting the outer membrane protein LptD (involved in LPS-translocation), is currently in Phase I clinical trials. A recent report also describes Dynobactin, a peptide antibiotic targeting the BamA outer membrane protein insertion machinery, as an effective new antibiotic [6]. The latter two examples are the most recent promising antibacterial compounds and exemplify the need to focus scientific efforts on novel strategies, especially in the fight against Gram-negative bacteria. 

In addition to possible acquired resistance mechanisms, Gram-negative bacteria already have a high intrinsic resistance to most clinical antibiotics, a property that can essentially be attributed to the combination of an additional outer membrane (OM) and the presence of powerful multidrug efflux pumps [7,8]. The highly asymmetric OM of Gram-negative bacteria, which is formed by lipopolysaccharides (LPS) on the outer leaflet and phospholipids on its inner leaflet, represents a significant permeability barrier, particularly for hydrophobic compounds such as bile salts, disinfectants, and most antimicrobials. Consequently, the OM reduces the uptake of antibiotics. However, as a passive barrier alone, it cannot influence the resulting intracellular equilibrium concentrations. Multidrug efflux transporters actively counteract influx across the outer (and inner) membrane. As a result, many antibiotics reach only sublethal concentrations at their sites of action within the bacterium [9]. Not surprisingly, multidrug efflux pumps have been found overexpressed in many clinical isolates [10,11]. As indicated by recent kinetic modeling studies and experiments, the interplay between slow uptake and active efflux appears to be finely tuned. Therefore, even small changes in one or both factors can dramatically increase intracellular drug concentrations, rendering the bacteria susceptible to antibiotic treatment [12,13]. This results in various possible approaches such as (i) the optimization of drugs for better influx and/or efflux avoidance [14,15,16], (ii) the permeabilization of the OM by additional chemosensitizers [2], or (iii) the inhibition of multidrug efflux pumps [17,18,19]. Synergistic approaches between (ii) membrane permeabilizers and (iii) efflux pump inhibitors (EPIs) can also be used to sensitize Gram-negative bacteria to antibiotics [20]. The inhibitory strategy appears to be the most versatile, since many antibiotics, including compounds that are ineffective per se for the reasons mentioned, can be (re-)used effectively against Gram-negative bacteria. 

## 2. Tripartite RND Multidrug Efflux Pumps

Members of the tripartite Resistance Nodulation cell Division (RND) superfamily are the major multidrug efflux pumps in Gram-negative bacteria [21]. Tripartite RND efflux pumps consist of membrane fusion proteins (MFPs, also known as periplasmic adaptor proteins, PAPs), an RND core component, and an outer membrane factor (OMF), which together form an elongated complex that connects both the inner and outer bacterial membrane. The RND core component is present in the inner membrane (IM) as a homo- or heterotrimer. RND proteins recognize drug substrates and energize the drug efflux at the expense of the proton motive force (PMF). The MFPs build a hexameric ring on top of the RND proteins. In the active complex, this MFP forms a tubular structure, which connects to the open OMF porin in the outer membrane. As a result, substrates can be taken up by the RND core from the periplasm (or the outer leaflet of the IM) and removed from the cell by extrusion through the long MFP-OMF conduit across the OM [22,23,24,25] (see also [7] and [8] for reviews). Some pathogenic Gram-negative bacteria contain multiple clinically relevant tripartite RND efflux pumps with partially overlapping substrate specificities such as MexAB-OprM, MexCD-OprJ, MexEF-OprN, and MexXY-OprM from *Pseudomonas aeruginosa* or AdeABC, AdeFGH, and AdeIJK from *Acinetobacter baumannii* that are either constitutively expressed (MexAB and AdeIJK), induced by stress, or overexpressed due to mutation [26,27,28]. In other Gram-negatives such as *Escherichia coli* (AcrAB-TolC), *Salmonella enterica* (AcrAB-TolC), *Klebsiella pneumoniae* (AcrAB-TolC), *Campylobacter spp.* (CmeABC), and *Neisseria gonorrhoeae* (MtrCDE), single RND-tripartite systems appear to be dominant [28].

The value of RND efflux pumps as molecular targets is underscored by their propensity to support the acquisition of additional resistance mechanisms [29,30], their overexpression in dormant bacteria [31], and their role in cell-to-cell communication [32,33], biofilm formation [34], pathogenicity, and virulence [35]. Strategies against multidrug efflux pumps have been targeting their (a) expression [36], (b) complex assembly [37], or (c) the fully assembled active transporter.

This review focuses on synthetic RND efflux pump inhibitors that target the RND core component MexB or AcrB, or the MFP AcrA. A total of eight inhibitor classes with validated target-specific effects are described for their (i) structure–activity relationships, (ii) chemical synthesis, (iii) mode of action, (iv) *in vitro* activity, (v) pharmacological properties, and (vi) *in vivo* activity. The binding mode of AcrB substrates and inhibitors is also analyzed comparatively.

## 3. Efflux Pump Inhibitors

### 3.1. PAβN

PAβN (Phenylalanine-Arginine-β-Naphtylamide, **1**), also known as MC-207,110 (Figure 1), was discovered in 1999 by Microcide Pharmaceuticals and Daiichi Pharmaceutical Company as the first inhibitor of RND efflux pumps in Gram-negative bacteria. It was identified through the screening of approximately 200,000 synthetic and natural compounds in combination with a sub-inhibitory concentration of levofloxacin (an antibiotic substrate of RND efflux pumps), in order to find compounds that would potentiate the activity of this antibiotic on *Pseudomonas aeruginosa* [38,39,40]. This screening was performed using different engineered strains of *P. aeruginosa* that overexpressed either of the three main *P. aeruginosa* RND efflux pumps: MexAB-OprM (PAM1032), MexCD-OprJ (PAM1033), or MexEF-OprN (PAM1034) [41].

#### 3.1.1. Structure–Activity Relationships (SARs)

PAβN (**1**) is a low-molecular-weight dipeptidic compound composed of an L-phenylalanine (AA_1_) linked to an L-arginine (AA_2_) substituted by a naphthylamine (Cap) (Figure 1). 

Several structure–activity relationship (SAR) studies were performed in order to decrease the cytotoxicity and improve the potency and the ADME properties of PAβN (Figure 1) [39,42,43]. Amongst the more than 500 analogues that were synthesized and tested, the three main areas of modifications investigated were the nature of amino acids 1 and 2 and the cap moiety. Studies showed the following:

**Replacement of AA_1_ and AA_2_**: It was described that amino acids 1 and 2 needed to contain both an aromatic and a basic moiety, though the order could be inverted [39,42]. Replacement of L-phenylalanine with an L-homo-phenylalanine led to a 2-fold improvement in EPI potency. In addition, ornithine or an aminomethylproline was accepted as an alternative basic amino side chain [39,43].**Substitution of the amide bond:** Methylation of the amide between AA_1_ and AA_2_ led to a slight improvement in compound potency and plasma stability [39].**Modification of the cap side:** Replacement of the naphthyl moiety by other fused rings such as 5-aminoindan and 6-aminoquinoline was tolerated [39]. However, the introduction of a 3-aminoquinoline moiety was preferred to reduce the cytotoxicity on mammalian cells *in vitro* [39].

These structure–activity relationship studies have allowed the identification of several compounds, such as **2**, **3**, **4** (MC-02,595), and **5** (MC-04,124) (Figure 2).

#### 3.1.2. Chemical Synthesis

PAβN and its analogues were synthesized by standard peptide solid phase synthesis (Figure 1) [44]. Starting from the carbonate resin **6**, which was synthesized in two steps, the first amino acid was grafted as a lithium salt. The lithium salt was then converted to tetrabutylammonium salt by reaction with tetrabutylammonium fluoride (TBAF). Intermediate **8** was then activated using 2,4-dinitro-1-fluorobenzene (**9**) and reacted with the second amino acid to form intermediate **11**. The Cap moiety was finally introduced after activation of the carboxylate with EDCI (1-ethyl-3-(3-dimethylaminopropyl)carbodiimide) and HOBt (hydroxybenzotriazole), and then the resin was removed under acidic conditions using trifluoroacetic acid (TFA) to release the final desired compound **13**.

#### 3.1.3. *In Vitro* Activity

PAβN inhibits the three RND pumps of *P. aeruginosa* (MexAB-OprM, MexCD-OprJ, and MexEF-OprN) and is also a substrate of these pumps, indicating it acts as a competitive inhibitor. It has broad-spectrum efflux pump inhibitory activity and is, therefore, able to potentiate the activity of different families of antibiotics such as fluoroquinolones, macrolides, oxazolidinones, chloramphenicol, and rifampin but not aminoglycosides [40,41].

PAβN was reported to have no intrinsic bacterial activity but PAβN can permeabilize the membrane of PAM2035 *P. aeruginosa* strains (deficient for MexAB-OprM efflux pump) due to its cationicity [45]. Indeed, being protonated at physiological pH, PAβN can displace divalent cations that bridge phosphate groups of OM LPS, leading to OM instability. In *P. aeruginosa* WT strains or strains overexpressing the three Mex RND efflux pumps, the minimum inhibitory concentration (MIC) of PAβN was 512 µg/mL [39,41]. At a concentration of 20 µg/mL, PAβN is able to boost the activity of levofloxacin on *P. aeruginosa* by 16-fold, and up to 64-fold on strains overexpressing the Mex RND efflux pumps (where basal susceptibility was lower). Analogues identified in the SAR studies (**2** and **3**, minimal potentiating concentration (MPC_8_) of 5 µg/mL [39]; **4** and **5**, MPC_8_ = 10 µg/mL [43]) showed similar efflux pump inhibition to that of PAβN (MPC_8_ = 10 µg/mL).

At a concentration of 25–50 µ g/mL, PAβN was found to also reduce the MICs of several β-lactam antibiotics (piperacillin, cefotaxime, ceftazidime, and ciprofloxacin) against *P. aeruginosa* WT by a factor of 2 to 4 (and by a factor up to 42 on *dacB* strains that overexpress AmpC) [45].

PAβN and its analogues were also shown to inhibit RND efflux in *Escherichia coli* [41,46], as well as the AdeFGH RND efflux pump of *Acinetobacter baumannii*, where it was described to boost the activity of clindamycin, chloramphenicol, and trimethoprim on different clinical isolates of *A. baumannii* (AB005, AB010, and AB014) [47].

#### 3.1.4. Pharmacological Properties

PAβN is unstable in murine serum, likely due to the cleavage of the peptide bond between AA_1_ and AA_2_ [42]. Analogue **4** (**MC-02,595**), where AA_1_ and AA_2_ are changed, had greatly improved stability in murine serum (95% recovery after 2 h). Similarly, compound **2,** where the nitrogen of the amide bond is methylated, was found to better resist cleavage by serum proteases.

PAβN and several analogues such as compound **4** were reported to be toxic for mice after intravenous administration (LD_50_ < 25 mg/kg, MLD (minimal dose causing lethality to ≥66% of the animals tested) <25 mg/kg) [43]. As mentioned previously, a basic side chain (such as ornithine) for AA_1_ or AA_2_ was found to be critical for effective efflux pump inhibition; however, this moiety was also identified as the major contributor to compound toxicity. Replacement of the ornithine moiety by an aminomethylproline (**5**, **MC-04,124**) allowed a 4-fold decrease in toxicity (MLD > 100 mg/kg) in rodents, while maintaining the same level of inhibition.

Pharmacokinetic profiling in rats [48] showed that in contrast to serum concentrations, a single dose of compound **5** led to high concentrations in several organs (kidneys, liver, and lungs) after 24 h. Similarly, repeated doses provoked a significant accumulation in these organs, and the accumulation in the kidneys led to nephrotoxicity [35].

#### 3.1.5. *In Vivo* Activity

Compound **2** was evaluated in a neutropenic mouse model of thigh infection with *P. aeruginosa* overexpressing MexAB-OprM (PAM1032) in combination with levofloxacin [39]. Immunosuppressed male CFW mice were infected by 10^5^ CFU of *P. aeruginosa* PAM 1032 intramuscularly and treated with levofloxacin subcutaneously at 30 mg/kg and compound **2** intraperitoneally at 30 mg/kg. At these doses, levofloxacin alone and compound **2** alone each resulted in bacterial growth similar to untreated mice. The combination compound **2**/levofloxacin led to a 3-log reduction in CFU after 4 h. Compound **5** (**MC-04,124**) was also evaluated in an *in vivo* model of infection in combination with levofloxacin and shown to be similarly effective [43]. 

During the development of this chemical series, it was shown that the basic moieties of PAβN and its analogues were essential for activity. However, these moieties are associated with unfavorable pharmacokinetic and toxicological profiles, explaining why the development of this chemical series was suspended [40]. As a consequence, these compounds are only used as research tools due to their broad-spectrum EPI activity to understand the role of efflux pumps in antibiotic resistance in *P. aeruginosa* and other Gram-negative bacteria.

### 3.2. NMP and Arylpiperazines

Arylpiperazines were first described by Bohnert and Kern in 2005 as being able to reverse multidrug resistance in *E. coli* (Figure 3). [49]. These compounds were identified through high-throughput screening of synthetic and commercially available *N*-heterocyclic compounds, on two *E. coli* strains overexpressing *acrAB* (3-AG100MKX) and *acrEF* (2-DC14PS), in the presence of levofloxacin (sub-inhibitory concentrations).

#### 3.2.1. Structure-Activity Relationships

Following the identification of the arylpiperazine series from the screening, different structure–activity relationship studies were performed [49]. It has been highlighted that the efflux pump inhibitory activity of these compounds depends on four parameters (Figure 3):

**The size of the linker** between the aromatic ring and the piperazine: Linker elongation led to an improvement in potency.**The substitution of the phenyl ring:** Substitution of the benzene ring with a halogen-containing moiety increased compound potency. In particular, the introduction of a trifluoromethyl group in the *meta* position was preferred (8-fold improvement in boosting activity).**The nature of the aromatic ring:** Replacement of the phenyl ring by a naphthyl ring led to a 5-fold improvement in potency.**The substitution of the piperazine**: Substitutions of the piperazine ring were not tolerated and led to decreased potency.

The SAR studies have allowed the discovery of NMP (**14**, 1-(1-naphtylmethyl)-piperazine) as one of the most potent arylpiperazines (levofloxacin-MRC_4_ = 50 µg/mL) (Figure 3) [49].

#### 3.2.2. Chemical Synthesis

NMP synthesis was described in three steps from 1-(chloromethyl)naphthalene **15** (Figure 2) [50]. The first step involved a Finkelstein reaction to generate the iodide derivative **16**, followed by a nucleophilic substitution with Boc-piperazine and cleavage of the Boc group in acidic media.

#### 3.2.3. *In Vitro* Activity

NMP by itself has no intrinsic antibacterial activity (MIC = 400 µg/mL) on *E. coli* strains overexpressing or deficient in RND efflux pumps [49].

At a concentration of 100 µg/mL, NMP is able to increase the intracellular accumulation of levofloxacin in *E. coli* strains overexpressing *acrAB* (3-AG100MKX) [49]. It has also been shown that NMP at 100 µg/mL reduces the MIC of levofloxacin by 16-fold on *E. coli* (2-DC14PS) overexpressing *acrEF* and by 8-fold on *E. coli* 3-AG100MKX.

NMP was also reported to boost the activity of several antibiotics such as oxacillin, rifampicin, chloramphenicol, various fluoroquinolones (levofloxacin, ciprofloxacin, norfloxacin, enoxacin, and pefloxacin), different macrolides (clarithromycin, erythromycin, and azithromycin), clindamycin, doxycycline, linezolid, and nitrofurantoin on *E. coli* overexpressing AcrAB-TolC [49]. For example, at 100 µg/mL, it reduced the MIC of oxacillin, rifampicin, chloramphenicol, and clarithromycin by 4- to 8-fold and the MIC of linezolid by 8- to 32-fold on *E. coli* strains (3-AG100MKX and 2-DC14PS) overexpressing efflux pumps. As expected, the addition of NMP has no effect on efflux-pump-deficient strains (such as 1-DC14PS), demonstrating that the boosting effect of NMP is due to the inhibition of the efflux pump. Furthermore, it has no effect on ketolides (telithromycin), glycopeptides, aminoglycosides, trimethoprim/sulfamethoxazole, and fosfomycin [49].

NMP was also tested on clinical isolates and showed moderate boosting activity on multi-drug resistant (MDR) Enterobacterales (*E. coli*, *E. aerogenes*, *K. pneumoniae*) [51,52] and restored partial susceptibility to fluoroquinolones particularly on *E. coli*. At a concentration of 100 µg/mL, it reduced the MIC of levofloxacin by 4-fold on more than 50% of fluoroquinolone-sensitive *E. coli* strains and more than 70% of the fluoroquinolone-resistant strains.

Noteworthily, NMP was able to partially reverse drug resistance in *A. baumannii* [53]. At 100 µg/mL, it reduced the MIC of several antibiotics (e.g., levofloxacin, ciprofloxacin, chloramphenicol, and linezolid) by 4-fold or more on the *A. baumannii* ATCC 1906 strain and on SB13 and U10247 clinical strains.

#### 3.2.4. Pharmacological Properties

The arylpiperazine backbone of NMP may be associated with a serotonergic agonist effect, rendering it potentially toxic for clinical use [54]. No *in vivo* experiment has been reported for the arylpiperazine series.

### 3.3. D13-9001

D13-9001 (**19**) was designed as a specific inhibitor of MexAB-OprM of *P. aeruginosa* and is characterized by a pyridopyrimidine core (Figure 4). This compound was obtained after optimization of an initial hit compound (**18**) identified by high-throughput screening.

D13-9001 binds to MexB and AcrB in the deep binding pocket region known as the hydrophobic trap and inhibits the efflux of drugs by MexAB-OprM and AcrAB-TolC, respectively [55].

#### 3.3.1. Structure–Activity Relationships

Evaluation of the hit compound **18** revealed that it was highly bound to plasma proteins, limiting its potency in the presence of serum [56]. This can be explained by the presence of a carboxylic acid function and the relative hydrophobicity of the compound. To solve this issue, the authors introduced polar moieties at position C4 of the thiazole ring or replaced the phenyl ring linked to the carboxylic acid. However, these modifications led to inactive compounds. The substitution of the styrene with a pyridine linked to an ether or ethylene group decreased the plasma protein binding (PPB), but also reduced the potency of the compounds [56]. 

Subsequently, Nakayama et al. considered a change in scaffold by introducing a quinolone, leading to ineffective compounds, or alternatively introducing a pyridopyrimidine, which resulted in a decrease in potency but greatly reduced PPB. In parallel, it was shown that the substitution of the carboxylic acid with a bioisostere such as a tetrazole improved the potency [57].

Next, the introduction of a substituent in position C2 of the pyridopyrimidine core was considered. Cyclic amines led to the most potent compounds, and substitution of those cyclic amines with alcohols or carbamates was tolerated. The tetrazole was spaced from the pyridopyrimidine core with olefins, leading to an increase in potency. However, only the *E* isomer displayed activity, and a photochemical isomerization was observed with ambient light [58]. To prevent this phenomenon, the electron-donating ethylene in position C7 was replaced by an electron-withdrawing amide. As a result, the *E*-isomer was stable after exposure to ambient light (*E*/*Z* ratio after 240 min: 99/1). Regarding the cyclic amine in position C2, a carbamate in the (*R*) configuration was revealed to be more potent than the corresponding (*S*) enantiomer [59].

Carbon-substituted analogues with aryl or alkyl moieties in position C2 were also evaluated, but this led to an increase in lipophilicity, resulting in a lower solubility and greater PPB [60,61].

To improve solubility, highly water-soluble moieties, such as quaternary ammonium salt, were added. This led to the identification of D13-9001 (**19**), a specific inhibitor of MexAB-OprM in *P. aeruginosa* [62] (Figure 5).

#### 3.3.2. Chemical Synthesis

D13-9001 was obtained in a 10-step synthesis starting with 2-acetamidopyridine-4-carboxylic acid (Figure 3). First, a coupling reaction was performed between 4-*tert*-butylthiazol-2-amine and the synthesized acyl chloride **21**. The aniline was then deprotected by the reaction with ethanol in acidic conditions. This allowed for the formation of the pyridopyrimidine core through the reaction of **23** with bis(2,4,6-trichlorophenyl)malonate (TCPM). Subsequent formylation by the Vilsmeier–Haack reagent formed using phosgene in DMF yielded the corresponding aldehyde intermediate **25**. The olefin derivative **26** was then synthesized by condensation with 2-(1*H*-tetrazol-5-yl)acetic acid protected by a PMB (*p*-methoxybenzyl) group, followed by decarboxylation. An aromatic nucleophilic substitution with (3R)-piperidin-3-ol hydrochloride was then performed after activation of the phenol with diphenyl phosphoryle chloride to yield intermediate **27**. Subsequent reaction with carbonyldiimidazole (CDI) and *N*′,*N*′-dimethylethane-1,2-diamine gave the corresponding carbamate **28**. The tertiary amine was then alkylated using *tert*-butyl-2-bromoacetate, leading to ammonium **29**. Finally, the tetrazole was deprotected by reaction with trifluoroacetic acid in anisole to yield D13-9001 (**19**) [62].

#### 3.3.3. *In Vitro* Activity

The SAR studies identified D13-9001 as a lead compound, characterized by a lower potency but improved physicochemical properties compared to the hit **18**, such as better solubility and reduced plasma protein binding. *In vitro*, a concentration of 2 µg/mL of D13-9001 decreased the MIC of levofloxacin and aztreonam 8-fold in a *P. aeruginosa* Δ*mexCD*Δ*oprJ*_Δ*mexEF*Δ*oprN* strain overexpressing MexAB-OprM [62]. The K_D_ values of D13-9001 binding to purified AcrB and MexB were 1.15 and 3.57 μM, respectively [50].

#### 3.3.4. Pharmacological Properties

In addition to its demonstrated *in vivo* efficacy, D13-9001 is very soluble (747 µg/mL at pH 6.8) due to the presence of the quaternary ammonium. It also shows a linear PK profile in rats and monkeys after IV administration. A plasmatic concentration of 7.64 µg/mL was measured after the intravenous administration of a 5 mg/kg dose. The half-life was comprised of between 0.18 h in rats and 0.41 h in monkeys. Moreover, D13-9001 exhibited a good safety profile in an acute toxicity assay [62]. Structural studies showing binding of D13-9001 to the hydrophobic trap of MexB and AcrB (discussed under Section 4.3) were followed up by multiple molecular dynamics studies [63].

#### 3.3.5. *In Vivo* Activity

The combination D13-9001/aztreonam was evaluated *in vivo* in an acute *P. aeruginosa* pulmonary infection rat model. The lowest dose of D13-9001 tested (1.25 mg/kg) in combination with 1000 mg/kg of aztreonam resulted in more than 62% survival after 7 days, whereas aztreonam alone led to only 12.5% survival, showing the efficacy of the designed EPI [62].

Since the publication of Yoshida et al. in 2007, no recent advances have been reported regarding the preclinical or clinical development of D13-9001.

### 3.4. MBX Compounds

Pyranopyridines were first described in 2014 by the US company MicroBiotix as AcrAB-TolC efflux pump inhibitors in *E. coli* [64]. Cell-based high-throughput screening of 183,400 small molecules led to the identification of MBX2319 (**30**) as a potentiator of ciprofloxacin in *E. coli*.

#### 3.4.1. Structure—Activity Relationships

MBX2319 (**30**) is composed of a pyridine core substituted by five groups (Figure 6). In order to improve the potency, solubility, and metabolic stability of this compound, structure–activity relationship studies have been performed, in which the different substituents have been modified (Figure 6) [65].

**Replacement of the nitrile group**: Only two examples were synthesized due to synthetic problems, one with an *N*-hydroxyamide and the other with an amide, but these modifications led to a decrease in potency (MPC_4_ (levofloxacin and piperacillin) > 100 µM).**The oxidation of the sulfide group** to sulfoxide or sulfone also led to a decrease in potency. **Modification of alkyl linker**: The deletion of the chain on the sulfide, the modification of the chain size (one or three carbons), and the oxidation of the chain led to a total loss of activity.**Modification of gem-dimethyl moiety**: Replacement of gem-dimethyl substituent by hydrogen atoms also led to a total loss of activity.**Modification of the morpholine ring**: The morpholine was replaced by a variety of acyclic or cyclic (5,6,7-membered rings) amine. In general, the replacement was tolerated, but the introduction of substituents with basic amines was found to be associated with moderate/strong cytotoxicity. Only the introduction of a 2,6-dimethyl group allowed a good balance between activity and cytotoxicity.**Substitution/modification of the phenyl ring**: The introduction of substituents on the phenyl ring was tolerated, and the activity of the compounds depended on the substituent in the *para* position. The introduction of a neutral or basic substituent on the phenyl ring seemed to improve the activity.

The SARs allowed the discovery of three compounds: MBX3132 (**31**), MBX3135 (**32**), and MBX4191 (**33**) (Figure 6) [65].

#### 3.4.2. Chemical Synthesis

The different pyranopyridines were synthesized via a three-step synthetic approach (Figure 4) [65]. It started with a three-component coupling between pyranone, malonitrile, and disulfite carbon to form the thione **35**. The reaction of this thione with cyclic amines led to the formation of different substituted dihydropyranopyridines **36**, which were then alkylated to produce the corresponding desired product **37**.

#### 3.4.3. *In Vitro* Activity

Pyranopyridine compounds did not exhibit membrane-disruptive properties and did not possess intrinsic antibacterial activity (MIC > 100 µg/mL) [64].

MBX2319 (**30**) potentiated the activity of several families of AcrB substrate antibiotics: Fluoroquinolones (ciprofloxacin, levofloxacin), β-lactams (piperacillin, oxacillin), and chloramphenicol [64]. Indeed, at 3.13 µM, it decreased the MIC of ciprofloxacin, levofloxacin, and piperacillin by a 2-, 4-, and 4-fold, respectively, on *E. coli* WT strain (AB1157) and by 4- to 8- fold on *E. coli* strains overexpressing efflux pumps (285 and 287). It does not boost antibiotics in efflux-deficient strains. 

Moreover, MBX2319 improved the antibacterial activity of levofloxacin and ciprofloxacin by 2- to 5- fold on several other Gram-negative bacteria: *Shigella flexneri*, *Enterobacter aerogenes*, *Salmonella enterica*, and *Klebsiella pneumoniae,* and presented a weak activity with *P. aeruginosa* strains [64].

MBX3132 (**31**, MPC_4_ of levofloxacin and piperacillin: 0.1 µM) and MBX3135 (**32**, MPC_4_ of levofloxacin and piperacillin: 0.1 and 0.05 µM) are 30-fold more potent against *E. coli* than MBX2319 (**30**, MPC_4_ levofloxacin and piperacillin: 3.1 µM) [65].

#### 3.4.4. Mode of Action

Co-crystal structures of pyranopyridines were obtained and showed that MBX2319 binds to the periplasmic domain of AcrB (AcrBper) in the hydrophobic trap in the AcrB T protomer [66]. A detailed description of the interaction of the inhibitor with AcrB is given below (Section 4.3).

#### 3.4.5. Pharmacological Properties

While the hit MBX2319 showed a very low solubility (12 µM) and very high instability in mouse liver microsome (MLM) and human liver microsome (HLM) (0% remaining after 1 h at 37 °C in the presence of NADPH), the SARs performed allowed researchers to identify compounds with better physico-chemical and pharmacokinetic properties [65]. Compounds with ionizable groups had better solubility (MBX4191: >100 µM [67]) than neutral analogues (50 µM for MBX3132, 25 µM for MBX3135). Moreover, the introduction of a dimethylmorpholinyl group led to more stable compounds in microsomes (49% in MLM and 100% in HLM remaining after 1 h for MBX3135, 97% in MLM remaining after 1 h for MBX4191 [65,67]).

MBX compounds showed moderate cytotoxicity on mammalian cells (HeLa) (CC_50_ > 100 µM for MBX2319, CC_50_ = 60.5 µM for MBX3132, CC_50_ = 62.4 µM for MBX3135, and CC_50_ = 47 µM for MBX4191) [65,67].

#### 3.4.6. *In Vivo* Activity

MBX4191 (**33**) showed good serum exposure after intraperitoneal and intravenous doses of 10 and 100 mg/kg in mice [68]. The compound exhibited favorable pharmacokinetic properties allowing its evaluation in a mouse model of sepsis. MBX4191 potentiated the activity of levofloxacin in mice infected with a drug-susceptible strain of *E. coli* and potentiated minocycline in mice infected with a minocycline-resistant strain of *K. pneumoniae* [67].

### 3.5. 2H-Benzo[h]chromene Series

**WK2** (**42**) is an AcrB inhibitor characterized by a 2*H*-benzo[h]chromene core developed by Wang et al. [69]. This compound is the result of several medicinal chemistry programs. The authors first selected the 2-naphtamide core following an in silico screening. This led to the identification of 4-isopentyloxy-2-naphthamide as a moderate AcrB inhibitor (compound **A3**, **38**) [70,71]. The naphthamide moiety was then replaced by a methoxy-2,3-naphthalimide ring, yielding compound **A5** (**39**) [72]. Finally, the combination of the naphthamide approach with WD6 (**41**) [73], itself originating from the optimization of NDGA (**40**, nordihydroguaiaretic acid) [74], led to the 2*H*-benzo[h]chromene derivatives (Figure 7).

#### 3.5.1. Structure–Activity Relationships

Firstly, the structure of the core was investigated (Figure 8). It was revealed that chromanone derivatives were less potent overall than the corresponding 2*H*-benzo[h]chromene derivatives. Then, two points of derivatization were defined around the 2*H*-benzo[h]chromene core: The R substituent in position C5 and the R’ substituent on the phenyl ring linked in position C7. Concerning position C5, small polar groups were more favorable for the potency of the compounds. This could be explained by a possible hydrogen bond network between the compound, a water molecule, and residue Gln176 of AcrB, as described in the docking study. 

Lastly, a methyl group or acrylamide as R’ substituent gave the most potent compounds. To note, *para* substitution was more favorable compared to no substitution or *ortho* substitution [69] (Figure 8).

#### 3.5.2. Chemical Synthesis

**WK2** (**42**) was obtained in nine steps (Figure 5). First, the phenol function of salicylaldehyde was protected with benzyl chloride. A Stobbe condensation with diethyl succinate allowed the synthesis of the corresponding α,β-unsaturated ester **45**. The formation of the second aromatic ring was performed in acetic anhydride under reflux conditions. Then, the acetyl group was removed with potassium carbonate in methanol, allowing the [3+3] cycloaddition with 3-methyl-2-butenal to form the benzo[h]chromene core of compound **48**. The oxadiazole ring was obtained after the reaction with hydrazine hydrate to form the hydrazide **49**, followed by triethyl orthoformate to close the ring. Catalytic hydrogenation allowed the deprotection of the phenol function and reduced the double bond to form the 2*H*-benzo[h]chromene core of compound **51**. Finally, this intermediate reacted with 4-methyl-benzyl halide and potassium carbonate to yield compound **WK2** (**42**) [69].

#### 3.5.3. *In Vitro* Activity

Resulting from the SARs study, compound **WK2** substituted by an oxadiazole ring in position C5 showed the best compromise between potency and the ability to boost several antibiotics. Indeed, **WK2** boosted the antibacterial activity of chloramphenicol, erythromycin, tetraphenylphosphonium (TPP), and levofloxacin more than 4-fold at 128 µg/mL. To note, **WK2** displayed no antibacterial activity on its own and did not influence the MIC of rifampicin. To further confirm that **WK2** directly targeted AcrB, it was checked that it had no permeabilizing effect on the IM or OM of Gram-negative bacteria at a concentration of 128 µg/mL by assessing the membrane potential using DiOC_2_(3) (3,3′-diethyloxacarbocyanine iodide, a fluorescent membrane potential probe) and measuring the rate of hydrolysis of nitrocefin (a chromogenic β-lactam). Finally, it was shown that **WK2** inhibited the efflux of Nile Red, a dye specifically effluxed by AcrB, with complete inhibitory activity at 100 µM [69].

#### 3.5.4. Pharmacological Properties

**WK2** showed low cytotoxicity on HepG2 cells (87% viability at 100 µM) [69]. No data were provided on the solubility of this compound; however, its high hydrophobicity may predict low aqueous solubility.

No *in vivo* activity of the 2*H*-benzo[h]chromene derivatives has been reported in the literature yet.

### 3.6. NSC Series

NSC-33353-derived compounds have been recently described as MFP AcrA inhibitors and were able to boost the activity of antibiotics such as novobiocin and erythromycin. Screening of 1593 compounds (NCS Diversity Set V) at the NCI and the Developmental Therapeutics Program (DTP) in *E. coli* WT led to the identification of seven AcrA inhibitors, including NSC-60339 (Figure 9) [37]. The binding mode of these compounds was more precisely investigated by Darzynkiewicz et al. [75].

#### 3.6.1. Structure–Activity Relationships

NSC-33353 (**52**) is a 4,6-diaminoquinoline linked at the C-6 position to a cinnamoyl amide (Figure 9). Previous results showed that the quinoline ring was important for the interaction with AcrA. In order to improve the potency of the compounds, structure–activity relationship studies were performed on the cinnamoyl moiety (Figure 10) [76]:

**Substitution of cinnamoyl moiety**: First, the substitution of the cinnamoyl group was explored. The introduction of other electron-withdrawing (Cl, Br, NO_2_) or electron-donating (isopropyl) groups in position C-2, C-3, or C-4 did not improve the boosting effect of erythromycin (MPC_4_ = 3.1–200 µM) and novobiocin (MPC_4_ = 6.25–400 µM). Furthermore, these modifications generally led to an increase in the intrinsic antibacterial activity of the compounds.**Replacement of cinnamoyl moiety by substituted naphthyl rings**: Because the cinnamoyl moiety was thought to be responsible for the high cytotoxicity of NCS-33353, its replacement with a substituted naphthyl was performed. The introduction of electron-withdrawing (Br, CN, CO_2_Me) or electron-donating (OMe) groups at various positions (C-5, C-6, C-7) allowed a slight decrease in cytotoxicity but did not improve the boosting effect of novobiocin and led to compounds no longer able to boost erythromycin. In order to improve the affinity of the inhibitor for AcrA, aromatic substituents were added in C-5 and C-6 positions, but this led to a decrease or even total loss of the boosting effect on erythromycin and novobiocin.

**Figure 10 antibiotics-12-00180-f010:**
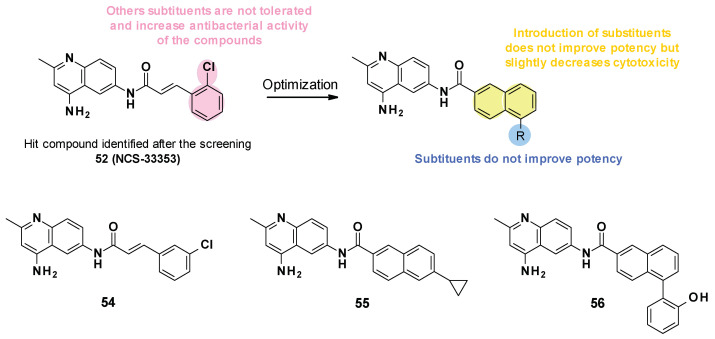
SARs summary and structures of compounds identified after SAR studies.

The SARs allowed the discovery of three compounds (Figure 10).

#### 3.6.2. Chemical Synthesis

NSC-33353 and its analogues were synthesized via a one-step procedure from commercially available 2-methylquinoline-4,6-diamine **57** (Figure 6) [76]. The synthesis consisted of a selective acylation in the presence of a substituted cinnamoyl chloride or naphthyl chloride. Compounds bearing phenyl on the naphthyl ring were obtained from bromosubstituted naphthyl analogues **60** by the Suzuki reaction.

#### 3.6.3. *In Vitro* Activity

NSC-33353 was able to potentiate the activity of novobiocin and erythromycin with an MPC_4_ of 1.56 and 3.1 µM on *E. coli* [76]. It was reported to have a low antibacterial activity (MIC = 200 µM on *E. coli* WT) but was a substrate of the AcrAB-TolC pump (MIC ∆*tolC* = 12.5 µM).

Cinnamoyl analogues showed higher antibacterial activity on *E. coli* WT (MIC = 25 µM for **54**) than naphthyl analogues (MIC = 200 and 50 µM for **55**, **56**) but also displayed a higher boosting effect of novobiocin and erythromycin than naphthyl analogues [76]. However, the boosting effect observed for compounds **55** and **56** remained less important than for NSC-33353. Noteworthily, NSC-33353 analogues permeabilized the outer membrane and were also substrates of the AcrAB-TolC efflux pump.

Furthermore, NSC-33353 potentiated the activity of erythromycin and novobiocin on other Enterobacterales such as *Enterobacter cloacae* (MPC_4_ of 12.5 µM for NOV and ERY) and *K. pneumoniae* (MPC_4_ of 12.5 µM for NOV). Conversely, no effect was observed on *P. aeruginosa* [76].

#### 3.6.4. Mode of Action

In surface plasmon resonance assays with purified AcrA, it was shown that all cinnamoyl analogues bound to AcrA with a K_D_ of 0.02 to 0.3 mM. In contrast, the naphthyl analogues bound to AcrA with different affinities (K_D_ of 0.001 to 1.18 mM) [76].

Docking studies showed that NSC-33535 bound to site III of AcrA within the AcrAB-TolC complex. This site is adjacent to AcrB and faces the AcrB AP [76].

#### 3.6.5. Pharmacological Properties

NSC-33353 showed significant cytotoxicity in A549 cells (CC_50_ = 10.0 µM) [76]. Naphthyl analogues, such as **55** and **56** (CC_50_ = 31.4, 27.3 µM, respectively), are slightly less cytotoxic than cinnamoyl analogues such as **54** (CC_50_ = 10.1 µM) and NSC-33353. No *in vivo* experiment was reported regarding the NCS series.

### 3.7. TXA Compounds

TXA09155 (**64**) is a compound developed by Taxis Pharmaceuticals to help treat MDR *P. aeruginosa* infections in burn victims or cystic fibrosis patients. They demonstrated its ability to inhibit the efflux of several antibiotics, including fluoroquinolones, tetracyclines, and β-lactams in *P. aeruginosa*.

TXA09155 is a conformationally constrained indole carboxamide derived from TXA01182 (**63**), itself originating from a synthetic screening of various heterocyclic carboxamide compounds stemming from aryl-alkyl diaminopentanamide EPIs such as **62** [77] (Figure 11).

#### 3.7.1. Structure–Activity Relationships

The compounds were designed not to cause membrane disruption, in order to avoid toxicity due to non-specific binding. The design was also focused on improving metabolic and serum stability compared to peptidomimetic EPIs. To do so, the aryl-alkyl moiety of the diaminopentanamide potentiators was first replaced with more druggable moieties such as fused heterocycles including benzothiazole, benzimidazole, benzofurane, benzothiophene, indole, and azaindole. These carboxamide-fused heterocycles were linked to a fluorobenzene on one side and a chiral diamine on the other side, yielding TXA01182 [78]. To enhance the potency and improve PK properties, the diamine side chain was conformationally constrained into a pyrrolidine ring, giving TXA09155 [79].

#### 3.7.2. Chemical Synthesis

TXA09155 (**64**) was synthesized in nine steps (Figure 7). First, commercially available (2*S*-4*R*)-4-hydroxypyrrolidine-2-carboxylic acid was allowed to react with thionyl chloride in methanol to form the corresponding methyl ester **66**. The secondary amine was then protected with benzyl bromide. The free alcohol was activated with tosyl chloride to perform a nucleophilic substitution with sodium cyanide. Subsequently, the methyl ester was reduced in primary alcohol by lithium borohydride. A Mitsunobu reaction allowed the introduction of the phtalimide, followed by its cleavage with hydrazine to give the corresponding primary amine **72**. This amine could then be involved in a coupling reaction with 6-(4-fluorophenyl)-1*H*-indole-2-carboxylic acid (**73**). The amide derivative finally underwent a reduction step and a deprotection step, followed by the addition of hydrochloric acid to yield TXA09155 [79].

#### 3.7.3. *In Vitro* Activity

Before assessing the potency of TXA09155 on various *P. aeruginosa* strains, its ability to inhibit the efflux of several antibiotics was confirmed. First, inner membrane permeabilization was measured using a flow cytometry-based propidium iodide assay and a nitrocefin assay was performed to monitor outer membrane permeabilization. In these two experiments, TXA09155 did not show significant membrane disruption at concentrations below 50 µg/mL. Then, it was demonstrated that TXA09155 inhibited the efflux of ethidium and levofloxacin. Lastly, TXA09155 did not affect the extent of the proton motive force and did not deplete ATP levels in *P. aeruginosa*, attesting that this compound is able to potentiate antibiotics by blocking their efflux.

Furthermore, the compound’s ability to boost antibiotics was tested on *P. aeruginosa* strains overexpressing MexAB-OprM or MexYZ-OprM, two RND-type efflux pumps. TXA09155 potentiated both cefpirome (third-generation cephalosporin) and levofloxacin by 8-fold. These results were confirmed on a panel of MDR *P. aeruginosa* clinical isolates: TXA09155 was able to lower the percentage of levofloxacin-resistant strains from 85% to 35%. Finally, TXA09155 reduced the frequency of resistance to levofloxacin by more than 149-fold, thus illustrating a further potential benefit of inhibiting efflux in bacteria [79].

Interestingly, the activity of co-trimoxazole, doxycycline, minocycline, and chloramphenicol was also potentiated by TXA09155 in Δ*oprM* strains.

The mode of action of TXA09155 was further explored with the selection of *P. aeruginosa* strains resistant to TXA09155 alone or to the combination of TXA09155/levofloxacin. Strains selected after incubation with TXA09155 alone possessed single-point mutations in the *phoQ* gene or a deletion in the *ompH* gene. *phoQ* codes for a component of the PhoP-PhoQ system, which regulates the expression of more than 100 genes in *P. aeruginosa*, including *mexX*. The deletion of *ompH* has been associated with hypersusceptibility to various antibiotics such as levofloxacin or doxycycline [80]. Hence, Zhang et al. hypothesized that OmpH could be implicated in the correct folding of efflux pumps [81], and in consequence, a deletion in *ompH* could provoke a misfolding of an efflux pump targeted by TXA09155 or TXA09155 could directly target OmpH, resulting in misfolded efflux pump components. On the other hand, strains selected after incubation with the combination of TXA09155/levofloxacine were characterized by a single point mutation in *trpS*, a gene coding for a tryptophan-tRNA ligase, further corroborating the implication of TXA09155 in protein synthesis inhibition as a secondary mode of action [79].

#### 3.7.4. Pharmacological Properties

TXA09155 showed favorable *in vitro* ADME properties: It is highly soluble (>155 µM at physiological pH) and relatively stable in human and rat liver microsomes. Nonetheless, it has shown inhibitory activity on cytochrome P450 3A4 (CYP3A4, IC_50_ = 28.5 µM) and hERG (IC_50_ = 16 µM), suggesting that this compound needs further optimization [79]. Indeed, CYP3A4 catalyzes reactions involved in drug metabolism, and its inhibition can lead to drug–drug interactions, while hERG inhibition can cause heart arrhythmias.

#### 3.7.5. *In Vivo* Activity

No *in vivo* experiment was published regarding TXA09155. However, a compound from the same chemical series (**75**, Figure 12) was tested in a murine septicemia model of infection by *P. aeruginosa*. The combination **75** (3.0 mg/mL bid)/cefepime (10 mg/mL) allowed 100% of the mice to survive at 72 h, whereas cefepime alone only showed 25% survival at 72 h [82].

### 3.8. Pyridylpiperazines (BDM Compounds)

Pyridylpiperazine compounds have been recently described as AcrB inhibitors and are able to boost antibiotic activity in *E. coli* [83]. BDM88855 (**78**) and derivatives act as allosteric efflux-pump inhibitors, which uniquely bind to the transmembrane domain of AcrB. This series was discovered by phenotypic screening of a 1280 fragment-library on *E. coli* BW25113 in combination with a sub-active dose of pyridomycin, a compound particularly effluxed by AcrAB-TolC. 

#### 3.8.1. Structure-Activity Relationships

BDM73185 (**76**), the hit compound identified from the fragment screening, is a small molecule with a pyridine core substituted by two electron-withdrawing substituents (Cl and CF_3_) in C-3 and C-5 positions and a basic piperazine moiety in the C-2 position (Figure 13). To improve the potency of BDM73185, structure–activity relationships were performed on the piperazine ring, the trifluoromethyl group, the pyridine moiety, and the chlorine atom [83]:

**Modification of the piperazine moiety:** Replacement by morpholine or piperidine led to a total loss of activity (EC_90_ > 500 μM) suggesting that the basic nitrogen was important for potency. In addition, the substitution of the amine with methyl led to a 6-fold decrease in potency.**Replacement of the trifluoromethyl group**: Substitution with a more polar group (OMe) led to a decrease in potency in contrast to the introduction of hydrophobic substituents. The introduction of halogen atoms was preferred and led to a 5-fold improvement of potency for the compound bearing an iodine atom (**77**, EC_90_ = 12 μM).**Replacement of the pyridine core**: Replacement with a quinoline core led to a 15-fold more potent compound (**78**, EC_90_ = 3.4 μM).**Replacement of the chlorine atom**: Removal of the chlorine atom led to a significant decrease in potency (EC_90_ > 250 μM), while replacement with other halogen atoms (Br, I) led to a slight improvement in potency (EC_90_ = 1.5 μM and 3 μM, respectively).

**Figure 13 antibiotics-12-00180-f013:**
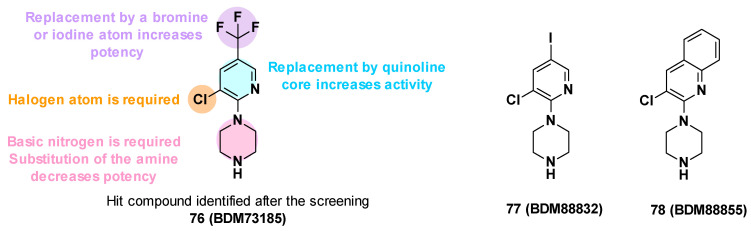
Summary of the SARs performed on BDM73185 (**76**) and structures of BDM88832 (**77**) and BDM88855 (**78**).

The SARs allowed the identification of different compounds such as BDM88832 (**77**) and BDM88855 (**78**), which were 5 to 18 times more potent than the hit BDM73185 (Figure 13) [83].

#### 3.8.2. Chemical Synthesis

BDM88855 (**78**) was synthesized in two steps starting from 2,3-dichloroquinoline (Figure 8) [83]. Firstly, aromatic nucleophilic substitution was performed with Boc-piperazine. The Boc group was then cleaved by hydrochloric acid in dioxane to afford BDM88855.

#### 3.8.3. *In Vitro* Activity

Pyridylpiperazine compounds did not exhibit antibacterial activity (MIC_90_ > 250 µM) [83].

At 300 µM, BDM73185 boosted the antibiotic activity of a panel of AcrAB-TolC substrates such as chloramphenicol (8-fold), pyridomycin (4-fold), tetracycline (4-fold), erythromycin, and ciprofloxacin (2-fold), but no effect was observed with non-AcrB substrates such as streptomycin and kanamycin. Moreover, it had no boosting effect on *E. coli* Δ*acrB*, Δ*acrA*, or Δ*tolC* mutants [83].

BDM88855.HCl (100 µM) was also able to boost all the AcrB substrates antibiotics: Oxacillin (>30-fold), linezolid (>30-fold), novobiocin (8-fold), and fusidic acid (8-fold) in *E. coli* wild-type, to levels similar to that observed for *E. coli* Δ*acrAB* [83].

#### 3.8.4. Mode of Action

To confirm the mode of action, resistant mutants were selected on *E. coli* in the presence of BDM73185 and erythromycin. Four strains resistant to BDM73185 but not to erythromycin were identified. Whole-genome sequencing revealed a mutation in the *acrB* gene, more specifically in the part coding for transmembrane helix 5, confirming the mechanism of action. The binding site of these compounds was further confirmed with the co-crystal structures of pyridylpiperazine-based inhibitors that were obtained and showed that BDM88855 acts as an allosteric inhibitor binding to the AcrB transmembrane domain (see Section 4.4) [83].

#### 3.8.5. Pharmacological Properties

BDM73185 and BDM88855.HCl showed no cytotoxicity on BALB/3T3 cells (CC_50_ > 100 μM) and have high aqueous solubility (>200 μM in PBS) [83]. To date, no *in vivo* experiment has been reported for pyridylpiperazines.

## 4. Structural Perspective on AcrB Substrate and Inhibitor Binding

### 4.1. E. coli AcrAB-TolC, the Well-Studied Model System

*E. coli* AcrAB-TolC is structurally and functionally arguably the best-studied tripartite RND multidrug efflux pump and transports an extraordinarily broad range of substrates [8,21]. Recently, the functional dominance of the constitutively expressed transporter was impressively demonstrated against the background of an efflux-deficient strain in which a total of 35 efflux pumps were deleted [84]. The structure of the AcrAB-TolC complex was elucidated via single-particle cryoelectron microscopy in an asymmetric conformation that presumably represents the active state of the efflux pump (PDB entry: 5O66, [23]). The asymmetric state in which each of the three AcrB protomers adopts a different conformation has been observed in numerous X-ray structures of the AcrB wildtype protein (or variants) and in complex with substrates or inhibitors ([7,8], this review).

Each AcrB protomer can be structurally divided into a funnel domain (FD) important for AcrB trimerization [85,86], a transmembrane domain (TMD) that energizes drug transport under PMF consumption, and a porter domain (PD) that contains the access pocket (AP) and the deep binding pocket (DBP) and mediates substrate reception, recognition, and translocation (see Figure 14a). The individual protomer conformations of AcrB represent different intermediates of a functional rotation cycle, which, in analogy to the F_1_F_o_-ATPase or/and based on their function, are called loose (L), tight (T), and open (O), or access, binding, and extrusion [87,88]. During functional rotation, the conformation of the FD remains the same, while the PD and the TMD undergo major conformational changes in a coordinated fashion [87,88,89,90]. As can be seen by the comparative co-structural analysis in Section 4.2, most substrates either bind to the AP of the L protomer or the DBP of the T protomer. While the AP (between PD subdomains PC1 and PC2) is present in both the L and T conformation, the DBP (between PD subdomains PC1, PN1, and PN2) is exclusively open in the T state (see Figure 14e). The two polyspecific substrate pockets are separated by the switch loop, a small, adaptive module that participates in substrate binding to both pockets [91,92]. In addition to the large AP and DBP, several transient channels have been reported to be involved in the supply of substrates toward these binding pockets. Both channel 1 (CH1), which originates from the TM7/TM8 groove of an L or T intermediate, and CH2 at the lower cleft entrance lead to the AP [89,91,92,93]. CH3 and CH4, on the other hand, connect the DBP with the central cavity and the TM1/TM2 groove, respectively [94,95]. All pockets and entry channels are closed in the O protomer but this protomer comprises a unique exit gate to allow substrates to leave via the AcrA/TolC conduit.

Within the TMD, the O conformation represents the occluded state in which D408 (and D407) is (are) protonated [96,97]. The TMD consists of 12 TM helices from which the first bundle (TM1-6) connects to PD subdomains PN1 (TM1) and PN2 (TM2) and the second bundle (TM7-12) is connected to the PC1 (TM7) and PC2 (TM8) subdomains. Residues D407, D408 (both TM4), K940 (TM10), and R971 (TM11) in the TMD core are involved in sequential protonation and deprotonation and are central to the energy conversion during functional rotation [8,96]. The transition from the O to the L state opens a water-filled channel on the cytoplasmic side through which the deprotonation of the proton relay network has been postulated [96]. Within the PD, the L state corresponds to a substrate-accessible resting state [23,98,99]. The L-to-T transition is associated with major conformational changes in both the PD and the TMD. Tight substrate binding within the DBP stabilizes the T state, allowing protons from the periplasm to enter the TMD towards the titratable residues D407 and D408. This protonation changes the electrostatics within the TMD and induces the T to O transition, leading to the extrusion of substrates [96,97].

### 4.2. Substrate Binding

Due to the small number of directly comparable co-structures for other RND-transporters, the comparative structural analysis carried out here is limited to *E. coli* AcrB. A comparison of AcrB substrates reveals a variety of physicochemical properties. The distinction between substrates, non-substrates, or inhibitors is therefore almost impossible on the basis of the differences in their chemical structures. AcrAB-TolC mediates resistance against compounds, which are very small and simple, such as n-hexane, or very large and complex, such as bleomycin [100]. However, dianionic β-lactams [101] and aminoglycoside antibiotics [102] are giving rise to poor or no resistance phenotypes in *E. coli*, respectively. As a common property, substrates and inhibitors of multidrug efflux pumps such as AcrB comprise one or more hydrophobic (often aromatic) moieties [103,104,105].

Known *E. coli* AcrB wildtype/substrate and inhibitor co-structures were analyzed for potential protein ligand interactions. All residues within a radius of 3.5 Å around the ligand including its hydrogen atoms were recorded in the PDB structures and assigned to the residues of AcrB. Emerging patterns are shown in Appendix A. It should be noted that within this simplified view, a relative positional relationship does not necessarily indicate a relevant biophysical interaction. Furthermore, the available high-resolution data indicate the structural importance of structural water molecules. More detailed information can be found in the respective publications [66,92,106]. However, even considering these limitations, some common binding patterns can be observed.

In the AcrB L state, molecules such as rifampicin (PDB entry: 3AOB and 3AOD, [91]), 3-formyl-rifamycin (6ZOB, [93]), and the two stacked doxorubicin molecules (4DX7, [92]) predominantly interact with the outer and inner access pocket (part of the entry tunnel CH2) and to a lesser extent with the AP-DBP interface towards the (closed) deep binding pocket (see Figure 14c and Appendix A). The macrolide antibiotic erythromycin apparently binds particularly strongly to the inner AP and the AP–DBP interface, but also has interactions with the DBP cave region (see Figure 14d). Another channel leading to the AP and DBP is CH1. It originates inside the TM7/TM8 groove. CH1 was suggested to be relevant for the uptake of smaller compounds such as β-lactams [93,107], phenicols, dodecyl β-D-maltoside (DDM), linezolid, and novobiocin [93]. DDM, a validated AcrB substrate, not only binds to the TM7/TM8 groove of AcrB [92,108], but has been recently co-crystallized in an L/T state in which DDM was bound to residues of the AP and the AP-DBP interface (6ZOA, [93]). This detergent was effectively clamped by residues of CH1 including I38, A39, P40, T463, F563, G675, Q865, E866, and S869 (see Figure 14g). The T protomer, in contrast to the L protomer, is characterized by its fully open DBP. In the T protomer, puromycin was found in an intermediate position interacting predominantly with the AP-DBP interface and the DBP cave but also with the DBP groove region and the AP (5NC5, [23]). Binding of fusidic acid, on the other hand, was more focused to the DBP cave and the AP-DBP interface region (see Figure 14h) (7B8S, [106]). In addition to the AP–DBP interface and the DBP cave region, rhodamine 6G (5ENS, [66]) and doxycycline (7B8R, [106], one out of two simultaneously bound molecules) share substantial contacts with the DBP groove. Levofloxacin (7B8T, [106]) and doxorubicin (4DX7, [92]) bind in the DBP cave and groove region while minocycline (2DRD [88], 3AOD, [91], 4DX5, [92] and 5ENT, [66]) and the second (of two simultaneously bound) doxycycline molecules (7B8R, [106]) interact almost exclusively with the DBP groove (see Figure 14i). In addition to co-structures of antibiotics to the periplasmic porter domain, fusidic acid co-structures (5JMN, [109] and 6ZOD, [93]) show simultaneous binding of three fusidic acid molecules to the T protomer transmembrane domain (TMD). One binding site is located at the TM1/TM2 groove at the entrance of CH4, and one molecule binds to the lower part of the TM7/TM8 groove at the entrance of CH1. A third molecule binds deeply into the TMD upon a conformational shift of TM11 and TM12 [93]. This co-structure with multiple bound fusidic acid molecules potentially represents an intermediate stage of transport and/or allosteric modulation sites.

### 4.3. Competitive Inhibitor Binding

Until the discovery of the BDM series of inhibitors, which bind to the TMD of AcrB (see below) [83], the only RND multidrug efflux pump inhibitors that could be structurally elucidated were bound to the DBP. As shown in Appendix A and Figure 14j,k, D13-9001 (3W9H, [55]) and the MBX series (MBX2329 (5ENO), MBX2931 (5ENP), MBX3132 (5ENQ), and MBX3135 (5ENR), [66]) essentially interact with the entire hydrophobic interaction face of the DBP. These include the DBP cave residues F136, Y327, and F628, as well as the groove residues F178 I277, A279, F610, V612, and F615. Consequently, inhibitor binding appears to differ from substrate binding in terms of the intensity and multiplicity of interactions. The interactions with the DBP region known as the hydrophobic trap (HT) include intense π-π stacking between the planar inhibitor backbones and the aromatic side chains of F178, F615, and F628 [66]. The hydrophobic (and aromatic) character of this DBP region also appears to be present in other (major) RND multidrug efflux pumps [105]. Inhibitors of the MBX series are also in contact with the AP–DBP-interface (M573, F617). Of note, the interactions are extended by the hydrogen bonding interaction of the inhibitors with the side chains Q151, S155, and Q176 and the carbonyl backbone oxygen of A286, which all coordinate water molecules, and in contact with the MBX inhibitor [66].

As a common mode of competitive inhibition, it is assumed that high-affinity interactions between the inhibitor molecules and the DBP prevent substrate binding via steric hinderance. In addition, inhibitor binding to the T state likely prevents the T-O transition, which presumably arrests the conformation change of the other protomers [66]. The latter assumption is supported by data on structural dynamics of AcrB in the presence of PAβN [110]. In principle, docking and molecular dynamic simulations indicate a similar mode of action for further inhibitors acting in the region of the substrate binding pockets [110,111,112,113,114,115]. Potent efflux pump inhibitors are therefore likely to fit perfectly into the target binding pocket, which also involves strong interactions with the HT. The simple fact that substrates as such are not permanently bound but are actively transported [116] implies that compared to inhibitor binding, substrate recognition is significantly more flexible and dynamic. Moreover, substitutional analysis of side chains inside binding pockets has revealed substantial resilience of the binding pockets upon change including retaining wildtype resistance phenotypes [8,91,93,106,117]. It appears obvious that the distinction between a substrate and a competitive inhibitor is gradual and strongly dependent on the properties of the respective efflux pump(s). For example, berberine, an excellent substrate of *E. coli* AcrAB-TolC [100], inhibits the MexXY-OprM efflux pump from *P. aeruginosa* [118]. Therefore, competitive inhibition in the case of polyspecific efflux pumps remains a complicated field of research, and alternative efflux pump inhibitors with an allosteric mode of action have been recently reported as a promising avenue for the inactivation of RND efflux pumps [83].

### 4.4. Allosteric Inhibitors Affecting Proton Coupling

The pyridylpiperazine-based inhibitor BDM88855 was resolved within the AcrB transmembrane domain bound to an (partially) induced binding pocket, close to the titratable residues D407, D408, K940, and R971, which are responsible for the translocation of protons, the essential second substrate of RND multidrug efflux pumps [83,119,120,121]. In the apo protein L state (but not in the T or O state), the titratable residues D407 and D408 within the TMD core are accessible from the cytoplasm [92]. However, the resulting water-filled pocket is not large enough to accommodate the BDM88855 molecule (see Figure 15b). Instead, BDM88855 binds to an alternative L state, in which the transmembrane helices, particularly those of the second repeat (TMs 7-12), are shifted, thereby enlarging the cytoplasmic binding pocket. The situation within the TMD might represent an L-T intermediate (see Figure 15b,c), The periplasmic domain, however, remains in the native L state (see Figure 15a). Accordingly, BDM88855 binding might arrest the L-T transition, which is believed to be a vital process involving major TMD rearrangements, at an early stage [83].

From a structural point of view, the analogy with inhibitor binding to the mycolic acid transporter MmpL3 from *M. tuberculosis* is striking. Although the structure and function of the periplasmic domains of AcrB and MmpL3 differ greatly, their homologous TMDs allow for structural comparison. Monomeric MmpL3 was crystallized in the presence of several anti-tuberculosis MmpL3 inhibitors, all bound to the (putative) proton relay network. The nature and position of the residues involved in this network show a strong analogy to those in AcrB (see Figure 16a). Similar shifts within the MmpL3 TMD between the unbound and the inhibitor-bound states suggest a comparable mode of inhibition for AcrB inhibitor BDM88855 and the MmpL3 inhibitors SQ109 (6AJG), AU1235 (6AJH), ICA38 (6AJJ) [122], NITD-349 (7C2M), and SPIRO (7C2N) [123], (see Figure 16b,c). Interestingly, rimonabant (6AJI) binds to a conformation that essentially corresponds to the native (6AJF, [122]) reference structure of MmpL3, and therefore seems not to induce the TMD shifts as seen in the co-structures for the other MmpL3 inhibitors.

From a structural point of view, considering the high level of conservation within the TMD centers of RND multidrug efflux pumps [105], compounds such as BDM88855 appear to be a versatile alternative or complement to “classical” DBP-based inhibition approaches. In fact, combining approaches could be a more sustainable solution, as both DBP- and TMD-based inhibition strategies on their own appear vulnerable to single amino acid alterations [55,83,124,125].

## 5. Conclusions

Efflux pump inhibitors have the potential to greatly assist in the fight against antimicrobial drug resistance. Indeed, the multiple roles of efflux pumps in bacterial resistance (antibiotics efflux, biofilm formation, cell-to-cell communication, pathogenicity, and virulence [32,33,34,35]) make them a valuable molecular target.

In this review, we provided an overview of the development of eight classes of EPIs, each able to boost one or more antibiotics in Gram-negative bacteria, from their discovery to their biological evaluation, as well as their mechanisms of inhibition and binding mode based on co-crystallographic structures and molecular dynamics simulations. The increasing chemical diversity of these EPIs, coupled with their function through the exploitation of different binding pockets, are very promising developments that will increase the chances of bringing EPI to the clinical phase. There are scarce *in vivo* data on the usefulness of the described EPIs, which can be explained by toxicity issues (cell toxicity for PAβN [40], off-target issue for NMP [54]), or a limited range of *in vivo* potency (MBX compounds [67], D13-9001 [62]). EPIs are usually active against one or more efflux pumps from different Enterobacteriaceae (PaβN, NMP, NCS, D13-9001, MBX, BDM), whereas some also have an extended activity spectrum toward *P. aeruginosa* and/or *A. baumannii* (PaβN, NMP, D13-9001, and TXA). *in vivo* data on the EPI resilience toward mutations that render the efflux pumps active but insensitive to EPIs are not documented.

Additionally, the question of clinical bacterial strains characterized by multiple resistance mechanisms (efflux pumps and expression of β-lactamases for example) still needs to be addressed. Regarding this co-occurrence of multidrug efflux and another specific resistance mechanism, it would be conceivable to test antibiotics/EPIs combinations in routine antibiotic susceptibility testing.

Nonetheless, the path is free for a structure-based approach on the basis of efflux pump/EPI co-structures, to develop further and more potent inhibitors. In addition, some EPI series are already well-advanced, such as TXA compounds [79], and new opportunities are arising with the development of allosteric EPIs, such as BDM88855 [83].

## Data Availability

Not applicable.

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
