# Peer review of "Update on the Discovery of Efflux Pump Inhibitors against Critical Priority Gram-Negative Bacteria"

_antibiotics, 2023, doi:10.3390/antibiotics12010180_

Round 1

Reviewer 1 Report

The review is interesting and well written. I only suggest a few minor corrections before publication:

Line 152: Write EDCI and HOBt with full chemical name upon their first mention in the text

Line 248: Error in reference formatting

Lines 317, 319, 320: E isomer

Line 321: (R) configuration

Line 322: (S) enantiomer

Scheme 4: Define PMB

Figure 6: change Remplacement to Replacement

Line 453, 596: Reference formatting error

Reviewer 3 Report

Article "Update on the discovery of efflux pump inhibitors against critical priority Gram-negative bacteria" is interesting but you need to update it more for vitro and in vitro activities you can go through these articles for additions like Role of medicinal plants from North Western Himalayas as an efflux pump inhibitor against MDR AcrAB-TolC Salmonella enterica serovar typhimurium: In vitro and In silico studies etc.
